# An Ethical and Societal Analysis for Biotechnological Methods in Plant Breeding

**Niels Louwaars** [1,*] and **Henk Jochemsen** [2]

1   Law Group Wageningen University, Wageningen University, Wageningen, and Plantum,
    Netherlands Seed Association, P.O. Box 9101, 6700 HB Gouda, The Netherlands
2   Christian Philosophy, Wageningen University, P.O. Box 9101, 6700 HB Wageningen, The Netherlands;
    hjochemsen52@gmail.com
*   Correspondence: niels.louwaars@wur.nl

**Abstract:** Technological developments in plant breeding, notably cisgenesis and gene editing, require a rethinking of biotechnology policies. In addition to legal debates about the definition of genetic modification in the Cartagena Protocol and at national and supra-national levels, and debates about the safety of the resulting products for mankind and environment, discussions are ongoing in society concerning ethical and societal questions. In this paper, we analyse the main ethical issues that need to be taken into account when evaluating contemporary plant breeding techniques. After a brief description of the state of the art in plant breeding, we discuss these main ethical issues. We take Consequentialist, Deontological and Virtue ethics as bases of our analysis. This results in a generally positive approach to gene editing, but also highlights several concerns, predominantly used by particular groups in society. This leads to a moral incentive toward transparency and options for operationalizing consumer choice.

**Keywords:** ethics; plant breeding; gene editing; integrity; social justice; freedom of choice; labeling

## 1. Introduction

Much of the current discussion about plant biotechnology relates to the legal and technical definitions of the concept of genetic modification and the questions around risk assessments arising from this legislation on products of various forms of plant biotechnology, notably cisgenesis and gene editing. In this paper, we intend to identify and clarify arguments that are raised beyond these legal and risk-related technical discourses. The integrated societal-ethical framework drawn up in 2002 for transgenesis [1] mentions different normative ethics approaches and different levels that conceptualize the discussions on transgenics in the late 1990s (Table 1 and [2,3]). Several are particularly relevant for medical applications; most also for crop biotechnology. These reports indicate that the possibility cannot be avoided that the list of ethically relevant aspects may remain incomplete. It furthermore concludes that weighing these is a political rather than a scientific (follow-up) question. They also indicate that the discussions about values and goals are conducted from different a priori visions about the permissibility of biotechnology as such. Despite that we consider that their framework with three basic ethical approaches: consequentialism, deontology and virtue ethics cover the main concerns, opportunities, and dilemmas also for the current wave of technologies in crop biotechnologies. When a basic ideology is unconditionally either in favor or against modern gene technology, there is no need to continue the discussion. In all other cases, boundaries may need to be drawn that require an analysis about which goals can justify that certain values are affected and to what extent.

**Table 1.** Levels and styles of ethics (adapted from [1]).

| Scale/Type of Ethics-Practice | Global | Macro | Meso | Micro |
|---|---|---|---|---|
| Consequentialism (proportionality) | Sustainability, Biodiversity | Scarcity, Consumer interests, Relevance | Commercial interests, Efficiency, Effectivity, Safety, Risk | Welfare, No-harm, Do good |
| Deontology (normative) | Dignity, Human rights | Fundamental rights, Access, Justice | Patenting, Integrity (organisms) | Consent (patients) Individual freedom for breeder/farmer Intrinsic value Integrity (object) |
| Virtue ethics (intentionality) | Justice | Citizenship Stewardship | Professionality Loyalty Image Responsibility Cultural identity | Attitude Integrity (subject) Care |

## 2. Societal Context

The societal context of the present discussion is on the one hand formed by the major goals of agriculture and plant breeding. Crop production is to support human needs toward food, feed, fiber, and many other products through the management of natural resources such as land, water, crop genetic resources. To sustain this into the future, this needs to be done with limited environmental impact, more circular with regard to material flows, and with a greater attention to biodiversity and soil management. At the same time, crop production must prepare for the effects of climate change, must continue to produce sufficient healthy food and animal feed (protein transition) for a growing and more affluent global population, and must also increasingly provide for plant-based raw materials for the 'greening' industry. Innovation in crop production is thus indispensable. Plant breeding has had a significant role in advancing agriculture to the needs in different periods of human civilization. It can now significantly contribute toward societal directions, such as the Green Deal of the European Union. On the other hand, it is understood that the urgency of realizing those goals make an acceleration of breeding processes highly desirable as conventional breeding takes a lot of time (between 6 and 20 years from crossing to seed of a new variety in the farmers' field).

Since the emergence of genetic modification in the 1980s debates have taken place in society about the ethical acceptance of biotechnology in animal breeding. Whereas welfare is a major issue in animal breeding, this plays a minor role in plants. There are no parallels in plant breeding to questions such as "Is the breeding of better meat producing cattle morally acceptable if the calves can only be born by Cesarean section?" Philosophical arguments about species integrity and intrinsic value of the plant do play a role though. Furthermore, questions are raised about how breeding contributes to or detracts from the type of agriculture that we might prefer, for example around herbicide-tolerant crops that enable large-scale cultivation, sometimes at the expense of the conservation of forest biodiversity. There is also public concern about the dominance of globally operating seed and agrochemical companies [4,5]. On the other hand, reference is made to the possibilities of the technology to help solve an ethically problematic situation. For example, the Ethics Council in Denmark [6] asks whether it would be unethical not to use technical possibilities in breeding when they may tackle morally important issues such as hunger and climate change. The technical possibilities thus become part of ethical dilemmas.

Before going into the ethical discussion in more detail, we will first give a general overview of the developments in breeding and the techniques that are currently under debate.

## 3. Breeding Methods

Plant breeding is basically the collection and creation of diversity, followed by selection within that diversity of plants that meet a predefined goal. In the last 150 years, various

insights in biological processes have been acquired that have enabled increasingly efficient and effective breeding. These methods either support the creation of diversity or facilitate selection. Three basic inventions laid the basis: the insight that the flower has male and / or female organs in the 17th century [7], which can be seen as the start of the conscious crossings creating diversity. In 1900 the principle of heredity was quantitatively explained on the basis of Mendel's papers [8,9], written some 35 years earlier, which lead to the selection of parents and descendants. This can be considered the start of scientific breeding. In 1953 the structure of DNA was published [10], which is the starting point of molecular genetics.

For the creation of "new" variation, the main breeding methods are:

- Introduction of diversity from other regions and crossing with the local genepool (from ca 1800 [11]);
- Use of hybrid vigor by creating hybrids using controlled crossing of selected parent lines (1925 [12]);
- Mutagenesis using ionizing radiation (1930s) [13,14]
- Duplication of the number of chromosomes with colchicine (1940s [15]);
- Chemical mutagenesis on plants / tissues and in cell cultures (1940s [16]);
- Embryo rescue, preventing abortion in crosses of related species (1950s [17]);
- Cell fusion and protoplast fusion with the aim of combining genomes (1970s [18]);
- Transgenesis: the transfer of functional genes from one species to another (1980s [19]);
- Cisgenesis: the transfer of functional genes within (or between crossable) species (2000s [20]);
- Targeted mutagenesis: the targeted cutting and/or replacement of base pairs (ZnFinger, TALEN [21], etc. 2005) and techniques based on CRISPR cas (2012 [22]).

Selecting new desired variants is done with the following methods:

- Line and family selection (1880s [23]);
- Mathematical statistics as an aid in the selection (1920s [24]);
- Tissue culture techniques for quick propagation which took off in the 1960s [25]
- Doubled haploids to accelerate homozygosity (1960s) [26];
- Molecular Marker-assisted selection (late 1980s [27]);
- Genomic selection (2010s [28]).

Until the introduction of transgenic plants and clones (sheep Dolly) and transgenic (bull Herman) farm animals in the 1980s, there never was a public debate about methods in plant breeding. A discussion about molecular techniques in the selection process quickly ceased. Greenpeace supported the use of molecular biological tools in selection processes [29], but not the use of molecular knowledge to create new variation. Ethical discussions about DNA-technologies arose in the health professions where genetic modification was broadly accepted in vaccine and drug development; debates are ongoing about affecting the germline though [30].

In plant breeding, such discussions have been sparked again over the past 15 years by the emergence of cisgenesis, as a potentially more acceptable alternative to transgenesis, and currently by the new gene editing techniques [31,32]. In some groups, notably in organic and biodynamic circles, this triggered discussions about older techniques as well, which on closer inspection are considered not to fit either within their views on agriculture. These techniques include forms of male sterility, duplication of chromosomes, mutation breeding and hybrid varieties.

*Gene Editing*

Gene editing (or genome editing) is the targeted modification of the DNA to insert, change or disable the function of a gene. Roughly three areas of application of these methods are distinguished (SDN = Site Directed Nucleases):

- In SDN-1 applications, mutations consisting of changes in a few base pairs, are generated as a result of an error-prone gene natural repair mechanisms of the DNA after double-stranded cuts are made at a particular in a predefined location.
- In SDN-2 applications, specific point mutations, small deletions / additions are generated as a result of the introduction into the cell of a repair DNA template. By means of homologous recombination (HR), precise and small genetic changes can be achieved in a gene adapting its function.
- In SDN-3 applications, entire functional genes can be inserted into a desired location in the genome through the delivery of exogenous donor DNA up to several kilobases long.

SDN-1 or SDN-2 techniques, leading to small changes in the genome without crossing species boundaries, do not result in plants that could not also be obtained with traditional—widely accepted—breeding methods or that could not arise spontaneously in nature. This is also the case with the site specific cis-genic replacement of a plant allele with another one from the same or a crossable species in an SDN-3 cis-genesis procedure. Transferring a gene from an unrelated species using SDN-3 produces plants that could not commonly arise in nature even though horizontal DNA transfer can occur from certain bacteria to their host plants, and it was even shown among (parasitic and ferns) plants [33]. Such transgenic approach will not be included in the further analysis.

## 4. Ethical Considerations—Consequentialism

Consequentialism or consequence ethics is an ethical-philosophical approach in which ethically correct action is characterized by a good result of the actions. In terms of plant breeding our focus is on the mainly utilitarian explanation of it. Initially that is, of course, to cater for today's human needs in terms of food, feed, fuel, and other products. However, plant breeding is a very future-focused enterprise, so in this section we will focus mainly on the (potential and probable) consequences with regard to sustainability, biodiversity, and potential negative effects derived from unintended side effects of the technology.

### 4.1. Sustainability

Sustainability is a general value; the concept was defined in the report Our Common Future, the so-called Brundtland report presented to the General Assembly of the United Nations in 1987 [34]. It attempts to do justice to the perspectives of People (social), Planet (ecological) and Profit (economic), and to find a balance between them. In terms of plant breeding, the question is to what extent breeding contributes to these components.

The 'People' aspect may, in addition to enabling socially resilient communities, relate to the contribution of breeding for health, i.e., through selection on health enhancing compounds such as vitamins, and bioavailable iron and zinc in food, or through making vegetables more readily available (and appealing) for more people.

Breeding on the one hand strongly focuses on yield enhancement and stability, and on the other on aspects of resource use efficiency, notably related to fertilizers and water, and on reducing food losses. Increasing the yield of a crop per surface area can be regarded as positive from a sustainability perspective (Planet and Profit), freeing up more space for nature and other land uses. Important yield determinants are resistant to diseases, pests and weeds, and tolerant to abiotic factors such as drought, salinity, heat, and cold. Further yield-related components include resource use efficiency (yield per amount of fertilization and water), harvest index (amount of usable product compared to less useful plant parts), and crop-specific issues such as reducing crop losses. Breeding for disease resistance can also limit the use of chemical crop protection and drought tolerance reduces the need for irrigation during a dry summer. Plant breeding can therefore make an important contribution to improving the 'Planet' aspects of sustainability when such breeding goals are prioritized.

Sustainability is also directly or indirectly related to product quality. This may concern processability (brewing quality of barley), shelf life of flowers, vegetables and fruit (reduction of food losses), and consumer qualities (taste, consistency, nutritional value).

The actual contributions of breeding to sustainability is under discussion. Breeding can adapt crops to certain cultivation techniques, whose contribution to sustainability is debated. An example is herbicide tolerance, which allows for large-scale cultivation because less labor is required for weeding, but which increases the use of chemicals with an impact on the environment. This can be considered negative, but in areas where erosion is lurking (dust-bowl in the US) zero tillage with herbicides, may be positive from a soil management point of view. Furthermore, although the most widely debated form ("Roundup Ready") of herbicide tolerance is a result of biotechnology, it can be pointed out that also non-GM plants exist that are tolerant to certain weed killers. It is, therefore, logical not to regulate herbicide tolerance through biotechnology laws, but rather through the crop protection regulations.

Policy choices can also have a marked effect in the 'Profit' component of sustainability, including unintended economic side effects at this level. The expenses needed to get a GMO cultivar accepted are very high due to the extensive regulation requirements. Due to this, GMO regulation-only internationally operating companies have the legal and financial capacity to deal with these regulations. Such regulatory costs can only be recouped when genetic modification is applied to the world's largest crops (especially corn, soy, and cotton), so the technology has almost exclusively been applied to those crops that are used for animal feed.

The profit component of sustainability is also illustrated by the protection of transgenic technologies and trait patents as opposed to protection of varieties through plant breeder's rights which has an important open innovation character. That aspect will be discussed under the heading of social justice.

### 4.2. Biodiversity

Biodiversity is a value and a policy goal linked to sustainability. It mainly concerns three system levels that are not often distinguished in societal discussions on biodiversity: diversity in ecosystems (landscape), species diversity within (agro-) ecosystems and genetic diversity within the species (within and between plant varieties). There are particularly some concerns about the genetic diversity resulting from plant breeding—the diversity between varieties from which the farmer can choose.

Historically, there have been three 'bottlenecks' to crop genetic diversity [35]: domestication (from wild plant to crop), distribution across the world (usually based on a small portion of the diversity in the areas of origin), and the innovation bottleneck when diverse farm varieties were replaced by high-yielding uniform varieties [36]. Little research has been conducted into trends in genetic diversity among cultivated plant varieties. Where this has been done (e.g., lettuce, wheat, tomato), it is clear that the genetic diversity in Europe is increasing as a result of modern breeding [37–39]. This is attributed to i) the introduction of breeder's rights, intensifying investment in breeding programs in the 1970s, ii) the development of marker-assisted selection in the 1990s, making genetically more distant material economically accessible for breeders (faster introgression—more efficient selection in back-cross programs), and iii) consumer demands, illustrated by the diversity of tomatoes in the 2010s [40].

However, concerns have been raised about the impact of modern varieties in the centers of diversity [41]. That impact may be small when traits enter the local genepool from commercial farmers' fields to landraces and crop relatives in nature that do not have a competitive advantage in such natural environments, such as herbicide tolerance. It could be, however, that a disease resistance gene resulting from scientific breeding might spread quickly in such populations together with associated traits from the modern variety, thus shifting the natural diversity. There is, however, no conceptual difference between mutant traits resulting from gene technologies or from conventional breeding.

Finally, for the genetic biodiversity level, it must be noted that breeders contribute to the conservation of genetic diversity in gene banks. Some gene banks are supported by private sector with monetary, but more often non-monetary contributions such as the multiplication and description of accessions. This Center for Genetic Resources in Wageningen thus saves a lot of labor and greenhouse space.

At the species diversity level in agro-ecosystems, breeding can in some cases increase the number of different crops grown in a certain area, for example through catch crops, the introduction of new (e.g., quinoa) crops or the (re-)introduction of crops, as is currently happening following European protein policy. Intensive breeding is needed to make European protein crops competitive with imported soy under current pricing. Breeding science also has a role to play in developing novel cultivation systems (e.g., strip cultivation), and enriching the microbiome in the soil. Plant breeding, therefore, plays a role in adapting crops to new—biodiverse—cultivation systems.

One aspect of biodiversity in which relatively little progress has been achieved is the breeding of genetically diverse varieties. Such diverse varieties can potentially provide a sustainability benefit at the field level. Genetically diverse farmers' varieties (landraces) disappeared during the time of agricultural mechanization and attempts in the 1970s to explicitly include diversity in 'multiline breeding' have been unsuccessful (wheat multiline TUMULT [42]). Gene editing might speed up the development of such multiline varieties [36]. Today, a concept of evolutionary breeding is taking root in discussions about sustainable farming systems and local products. This idea intends to lead to genetically diverse populations rather than clearly identifiable varieties that are allowed to evolve in their particular location and (commonly organic or biodynamic) farming system [43]. The actors in this field commonly reject the use of gene editing [44].

*4.3. Off-Target Effects*

When crossing plants to add a positive trait to a variety, it often happens that in the progeny, plants are found in which undesirable traits occur as well. In such cases, those plants are discarded in the selection process of breeding. This standard procedure of plant breeding, which also happens in plant domestication [45] does not create a moral dilemma similar to those in, for example, horse breeding, where killing most of the offspring of a cross is ethically unacceptable. It should be added that as a rule, conventional undirected techniques, for example random mutagenesis, lead to many more side effects or an unintended (lucky) positives, than targeted mutagenesis via CRISPR Cas. Such off-targets resulting from gene edits will be discarded in the same way as in conventional breeding [46].

It does happen, though, that certain intended goals come along with fewer ideal side effects that are connected. For example, cabbages such as Brussels sprouts that contain less glucosinolates (selected for a milder taste) are generally more susceptible to insect damage. In such cases it is up to the grower to balance (pest) risk and market opportunities of the tastier product.

Therefore, the discussion about such side effects in relation to gene editing are indeed very significant when applied to humans or animals, but the extension of concerns to plant breeding is not very logical. Discarding large numbers of plants in the breeding process at the most raises an economic, but not an ethical question.

*4.4. Social Justice*

A special type of side effects may be found in the field of social justice. New technologies not only have a direct effect on the crops and the growers, but they may also influence the context in which they are applied. Plant breeding has developed in an open innovation environment provided by plant breeder's rights. The 'breeder's exemption' allows anybody to freely use a protected variety for further breeding. This has supported a very diverse plant breeding sector.

During the period when plant biotechnology was emerging, a concentration in the seed sector took place, resulting in the establishment of large multinational companies that produce both GM-seeds and (the corresponding) crop protection chemicals. Along with the introduction of genetic modification, the patent system was introduced in the breeding sector, providing a much stronger right on both biotechnological methods and products (plants). As opposed the breeder's rights protected plant materials, a license is required to work with plants that include a patented trait. This has raised concern, at least in Europe, resulting in a restriction of the exclusive right of the patent holder (a 'limited breeder's exemption') in the Unitary Patent System. Furthermore, the European Patent Office decided that natural traits (products of essentially biological processes) are not patentable anymore. Next to these legal changes, FRAND (Fair, Reasonable and Non-Discriminatory) [47] licensing conditions, which should prevent 'strategic' use of the right [48], were established in the vegetable seeds sector and are debated elsewhere.

Intellectual Property Rights on plants had raised political debates since the 1980s leading in 2001 to the formulation of 'Farmers' Rights' [49] and, more recently, to debates about including Digital Sequence Information in the definition of 'genetic resource'. As a consequence, these 'genetic resources' would fall under the national sovereign rights of the countries where the physical genetic resources originate [50]. These rights-based debates are rooted in aspects of social justice. They hinge on regulatory systems that are not directly focusing on plant biotechnology but do impact their application in practice.

Social Justice in Hindsight

It is relevant to consider the effects of the policy decisions on transgenesis (GMOs) in the 1990s when evaluating the potential social justice effects of the technological developments in gene editing in the 2020s.

1.  The costs of the technique: Gene editing is technically simple and cheap to apply compared to transgenesis and is therefore potentially available to many breeders. "Service providers" have emerged to support smaller breeders and breeders of less commercial crops such as in floriculture, so that they do not have to invest in laboratories themselves.
2.  Regulation of biological safety: it costs (estimates vary) some 100 million Euro to bring one new GMO trait into the global market. This includes administrative and research costs to develop safety dossiers. This is one of the major reasons why only the largest companies have become active in the GMO market. Similar regulation of gene editing is likely to have a similar effect.
3.  Patents: whereas in the present European policy 'natural traits' of plants are not patentable, traits that have been created through technical means, such as products from gene editing, are protectable when they are sufficiently new, innovative and are usable, including gene edits. In addition a large number of process patents have been granted for the basic technology and for many further innovations/ applications in plant breeding. A breeder must find his way through that myriad of rights when he wants to use the technology, or to make a cross with an edited plant. The question is therefore whether such patents could bring about further shifts in the breeding sector, comparable to those following the introduction of transgenic GMOs.

## 5. Ethical Considerations—Deontology

Deontology is an ethical approach based on general principles and rules of conduct, commonly (but not always) set as norms. Here we mainly focus on rights (as legitimate moral claims and intrinsic value).

### 5.1. Rights

For people, we recognize the Universal Declaration on Human Rights [50]; for farm animals, the Animal Welfare Councils in The Netherlands and the UK [51,52] defined "five freedoms":

- Freedom from hunger and thirst;
- Freedom from discomfort;
- Freedom from pain, injury, and disease;
- Freedom to express natural behavior;
- Freedom from fear and distress.

In light of these rights various ethical dilemmas are being discussed related to techniques in which biotechnology could reduce animal diseases, preventing the large-scale killing of day-chicks, or cause discomfort by increased growth of certain body parts (meat cattle races that cannot give birth naturally; too heavy chicken, breaking their legs).

Parallel analyses are possible in relation to plant breeding. Lammerts van Bueren and Struik arrive at the following rights of crop plants from their organic / biodynamic point of view [53]:

(a) The right of the plant to fulfill its natural purpose and to be treated as an autonomous—self-regulating being (integrity of life);

(b) The right to complete the life cycle and to reproduce, given the agricultural ecology and the intrinsic biorhythms.

(c) The right to co-evolve with human development, but with respect for natural reproductive barriers (genotypic integrity);

(d) The right to be treated in such a way that the expression of the plant in form and function (phenotypic integrity) is consistent with the nature of the plant and human intentions.

These views have found little resonance in the scientific literature and in society outside (parts of) the organic sector. However, some of these do translate into the gene editing debate.

As regards (b) the right to complete the life cycle and to reproduce. This is not extended to the practice that many crops are harvested before they can reproduce (e.g., many vegetables) and others are grown because of their reproductive organs (grains) that are not used for reproduction. The argument was brought forward when a new principle (Genetic Use Restriction Technologies—GURTs) was applied to seed germination, resulting in crops whose grains could not germinate thus avoiding the reuse of farm-saved seed. This had the advantage for the seed company that the farmer has to buy fresh seeds for every cultivation cycle, and that possible cross fertilization with non-GM crops could be avoided. The resistance to this "Terminator Technology" has contributed to shelving it until now.

The same argument is used by a limited group within the organic sector to reject hybrids. Hybrids do produce offspring, but they is not genetically identical to the parents as with conventional varieties. Therefore, the plant can reproduce, but the plant variety cannot. Not accepting hybrids thus entails a broad interpretation of this right.

*5.2. Genotypic Integrity*

Within the organic sector, the principle of intrinsic value of plants has explicitly taken hold in the concept of "integrity of the cell" (or of the DNA), which is directly linked to breeding. Here, not the properties of the plant, but the activity and the actor [43] who directly intervenes in the cell and the DNA, creates the ethical issue.

From this point of view, a mutant that arises in nature (by cosmic radiation) is fully acceptable, but a plant with an identical DNA sequence that has arisen through human intervention (via targeted or random mutagenesis) is not. In this view, both transgenesis, cisgenesis and gene editing are incompatible with organic cultivation. Cytoplasmic male sterility resulting from cell fusion is also not acceptable, whereas such sterility found in nature is accepted, except for those that fundamentally reject the use of hybrids in which such traits are used. Doubling the ploidy level with natural colchicine (from the crocus—*Colchicum autumnale* L.) is permitted by some within the organic sector, but not when synthetic colchicine is used.

For some techniques, their use can be demonstrated unambiguously in the resulting material (e.g., male sterility), for others this is not possible. For example, it is impossible to determine whether a modern variety has an (induced) mutant in its family tree when the pedigree of a variety is not fully known. IFOAM is flexible about this: where it is known that such a mutation is in the variety, it is recommended not to use the variety, but extensive research into the pedigree is not considered necessary. The Biodynamic sector (Demeter) is also flexible about hybrids—these are not accepted in cereals, but vegetable hybrids are when insufficient good open-pollinated varieties are available [54].

### 5.3. Phenotypic Integrity

Phenotypic integrity is understood as a plant with the form and function corresponding to the nature of the plant. Discussion may arise when plants are used for the production of new (raw) materials for the biobased economy. Would the production of inositol in sugar beet, or pharmaceuticals in tobacco or banana compromise a phenotypical violation of integrity? Or does the production of rubber in Russian dandelion by Keygene [55] also fall under it, although it mainly concerns increasing the natural contents of these substances through breeding? Phenotypic integrity in a stricter sense poses major problems though. Our crops do hardly ever resemble the wild relatives from which they originate in form and function (just as a dachshund does not look like a wolf). It is not always clear where the boundary is set in the discussion about phenotypic integrity and its translation into plant rights as discussed earlier.

### 5.4. Intrinsic Value

In breeding, as in cultivation, we do things with plants in the interest of people. Especially since the Enlightenment, domestic animals and crops have primarily been assigned a utility value. Since the 19th century the awareness has slowly grown that an animal not only has a utility value but also an intrinsic value, meaning a value in itself apart from a possible utility for humans. In 1981 this concept was included in various Animals and Flora and Fauna Acts. In plant breeding and crop production, this recognition has not led to a recognition of the general protection of such values of cultivated plants though.

A corollary of this concept is the question as to when the intrinsic value of the individual is affected by a technique. Related to this is the concept of the integrity of the species. Notably from the philosophical perspective that nature is seen as creation, and that the order in living nature goes back to the creation of organisms "after their kind", an objection can be made to crossing species barriers. This view leads to the rejection of transgenesis in which functional properties of some organism (often a completely different species) are transferred to the plant with technical aids. In higher animals, the species concept is largely based on the inability to produce fertile offspring (e.g., mules), but that distinction is not as clear in the plant world. Related species can sometimes cross in nature or with minimal assistance (e.g., a bridge cross). This is used in conventional breeding to transfer useful properties between related species (e.g., clubroot resistance in cabbage (Brassica oleracea), originating from turnips (Brassica rapa). In the discussion about breeding methods, this concept is therefore mainly understood to mean the combination of (properties of) organisms that cannot cross.

Although transgenesis is rejected, cisgenesis may be accepted in this world view, since involves the transfer of functional genes between crossable species using biotechnology. Targeted mutagenesis would then also be acceptable, illustrated by the Dutch Government position stating The Netherlands support the application of techniques such as CRISPR Cas9, "provided that species boundaries are not crossed" [56]. The 2021 report by the European Commission on 'Novel Genomic Techniques' [57] also combines discussions of both cisgenesis and gene editing techniques. This is, however, mainly on the basis of safety rather than explicit ethical considerations.

## 6. Ethical Considerations—Virtue Ethics

Virtue ethics is a philosophical-ethical approach that answers the question: how should I live a (ethically) good life? This answer is formulated in terms of virtues, destiny (telos) and happiness, expressing virtues such as caring, responsibility, persistence, integrity. Related to plant breeding, virtue ethics indicates that the breeder has a moral responsibility and the result of his/her work will benefit others that need it.

### 6.1. Moral Responsibility and Principles

The starting point here is therefore that breeders, but also the parties in the value chain using the breeder's products have a (moral) responsibility that is not fulfilled by only adhering to regulations, but that requires behavior characterized by virtues. This can occur at an individual level, but also through sectoral action directives, such as sustainability labels. Taking corporate responsibility may concern environmental (Planet) but also social (People) matters such as child labor and living wages in production countries. The OECD has defined voluntary principles [58], but debates are ongoing to move from a soft-law approach [59] toward regulating at least the companies' due diligence in implementing such responsibilities [60,61] (If principles are established in regulations, virtues ethics has transitioned to deontology and law. However, to habitually observe such principles the professionals and the organizations and companies involved need to embody the corresponding virtues. Therefore, a regulation of biotechnology will only work well if attention is paid to these virtues). The organic sectors add additional moral principles and the corresponding virtues. Certain matters have been laid down in regulations, such as limiting chemical crop protection and chemical fertilizers, and rejection of regulated genetically modified crops (transgenics). In addition, specific parts of the sector have imposed additional restrictions, partly depending on the philosophy adhered to by the specific chain (such as biodynamic).

### 6.2. Transparency

Transparency in value chains can be considered a way to express care and professionalism of the actors. Transparency can help consumers to choose in two distinct ways: labeling and broader quality marks. Health related product qualities are commonly printed on labels, such as content (lipids, carbohydrates, and additives) of processed products, and information regarding allergens. Information about how the product was produced are commonly left to private quality marks, such as environmental (Utz / Rainforest Alliance for coffee); or social (Fair Trade, Slave-free) aspects of sustainability. Several marks combine aspects of both.

#### 6.2.1. Moral Responsibility of the Breeder

The breeder has the moral responsibility to inform the customers about the breeding methods used to create new varieties when asked. It is up to these customers whether and how to transfer that information down the value chain to finally reach consumers. This can be done through public databases or simply by informing customers that are part of a particular value chain about which varieties they can (and which cannot) use. Since different value chains may have very detailed wishes, which may even change over time, the latter mechanism would likely be most future-proof.

In such transparency measures, there always needs to be a disclaimer. A breeder cannot know all the breeding methods used to develop the parent materials that he uses to develop new varieties, when such materials originate from a country where editing is not regulated, or when edits have entered natural populations by crossing. Given that edits are currently scientific novelties that are published (and/or patented), this issue is not likely to emerge on a large scale. However, if editing becomes a household breeding method around the world, the extent of breeder's responsibility will become an issue. Since it cannot be assessed by testing the materials whether a plant was edited or whether the trait is a natural mutant, full guarantees cannot be given [62]. That is the case for techniques

such as mutagenesis through irradiation (in large-scale use since the 1950s) but also for gene editing. A breeder can be asked, however, to disclose the methods he used himself. Such transparency by the breeder and seed supplier may be translated in the value chains in different ways.

### 6.2.2. Labels on Consumer Products

Product labels would create the ultimate opportunities for consumers to choose among products. There are some challenges though with product labeling when it comes to breeding methods. As indicated above, several groups in society may accept different methods and not others. Those rejecting crossing species barriers only are served by the GMO-label for transgenics; others may reject hybrids, mutagenesis, gene editing or a combination of those; yet others may particularly reject herbicide tolerance irrespective of the breeding methods. Accommodating these different wishes on product labels could theoretically be done using modern tracking and tracing systems (blockchain) and QR-codes on both processed and fresh products. However, presenting on every product all breeding methods that have been used, puts a significant strain on processors (how to label a frozen pizza?) and the consumer (should he know the differences between TALEN, ODM, CRISPR Cas and CRISPR cpf, etc.)? That strain translates into higher costs for consumers which in itself may become an ethical issue in itself. So the question is to what extent such extensive information would be both possible and desirable?

### 6.2.3. Quality Marks

Similar to complex sustainability components, quality marks may strike a deal between clarity and simplicity for the consumer. Since organic label secures freedom from regulated GMO (next to the agronomic rules), the Biodynamic label puts additional restrictions with regard to breeding methods. It could well be that if there would be a particular consumer demand of products that are not made with gene editing, several existing brands may take it up in their certification schemes, or new marks may arise in the market.

### 6.2.4. Role of Government

Some decades ago government in several countries codified an obligatory GMO-label. In Europe, that label has not included so-called non-regulated GMOs such as products of mutation breeding and cell fusion. With the emergence of gene editing the regulator is challenged to decide whether such products should fall under the GMO-label; whether a new label is to be introduced or whether gene editing should not fall under a regulated label. The European Group on Ethics [30] considers that since a gene edit can often not be assessed in a final product, and since breeding used materials from different jurisdictions, an obligatory labeling requirement cannot be implemented. Hence, other means need to be designed to secure a freedom of choice for consumers, for example through inclusion in the organic label, and/or the creation of other private labels. Whatever the system, there will have to be a disclaimer. The same is likely to be the case with future technological developments.

## 7. Conclusions

Various considerations play a role when assessing the ethics around gene editing depending on the different inroads are taken. From a utilitarian/consequentialist point of view the expected positive contributions to environmental sustainability with regard to biodiversity and the need to reduce chemical crop protection and adapt crops to climate change appear significant. Gene editing and cis-genesis are primarily an accelerator toward such societal goals. Questions with regard to socio-economic aspects of sustainability point at avoiding overly expensive risk assessments and restrictive use of the patent system.

At the level of deontology, groups in society may oppose crossing species barriers or reject certain plant breeding methods based on views on integrity of the DNA, the cell, or the plant. Such aspects appear to weigh much more on applications of gene editing

in the germline of animals and humans. Analysis from the perspective of virtue ethics point particularly at moral responsibilities of the breeder and chain partners with regard to transparency. There is some logic to transfer such transparency to the consumer through quality marks rather than obligatory product labeling. In both cases, technical limitations have to be taken into account.

**Author Contributions:** This paper does not contain empirical research. For the whole paper both authors are responsible. For the parts on plant breeding technologies and developments: N.L.; for the ethical parts: H.J. All authors have read and agreed to the published version of the manuscript.

**Funding:** This research received no external funding.

**Institutional Review Board Statement:** Not applicable.

**Informed Consent Statement:** Not applicable.

**Data Availability Statement:** Not applicable.

**Conflicts of Interest:** The authors declare no conflict of interest.

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
