# Peer review of "An Ethical and Societal Analysis for Biotechnological Methods in Plant Breeding"

_agronomy, doi:10.3390/agronomy11061183_

Round 1

Reviewer 1 Report

Louwaars & Jochemsen discuss some ethical issues sorrounding contemporary plant breeding techniques. In my opinion, the paper fall short on several aspects:

1st) As the authors themselves recognize: "the discussions about values and goals are conducted from different a priori visions about the permissibility of biotechnology as such". The selection of ethical issues they decided to discuss/include is not clearly stated or presented, nor the reason behind this choice. Such issues are mentioned in Sect.2, but are not clearly discussed thereafter. In short, they fail to state first of all what is the purpose of agriculture and therefore of plant breeding. A reasonable consensus among professionals is that the primary purpose of agriculture is to produce food and other goods (feed, fiber, wood, flowers...) for human beings and, while doing that, take care of the land as to avoid irreversible damage that might compromise future use. Plant breeding would then serve as a mean to achieve such goals by adapting plant varieties to better fulfill human needs, maximising benefits and, at the same time, minimising damage. If this is true, the authors should always weight achievement in such goals against damage/risk.
If the authors have a different view of agriculture, and therefore of plant breeding, they should first of all make it clear and justify their choice. Indeed, in the introduction, the authors state they do not want to focus on the "[...] legal and technical definitions of the concept of genetic modification and the questions around risk assessments arising from this legislation." On the contrary, "...they intend to identify and clarify arguments that are raised outside these technical and legal discourse".
The outcome of such analysis (for instance in Consequentialism) depends a lot on the choice of what is considered a consequence and which ones are regarded as good. These must be clearly stated in the beginning, before adopting the different approaches.

2nd) Figures are of poor quality (Fig.1 is not clear or discussed in depth; some text is concealed or mispelled in Fig.2  ), are not really well explained or discussed in the text.
Fig. 1 comes from a paper in dutch, which is obviously not accessible to most readers.

3rd) Line 271-274 "The ‘breeder’s exemption’ allows anybody to freely use a protected variety for further breeding.... Along with the introduction of genetic modification, the patent system was introduced in the breeding sector, providing a much stronger right on both biotechnological methods and products"
This is unclear: do the authors claim patents are preventing further work by other subjects? More importantly, do they claim this a stronger obstacle than national regulations and Cartgena protocol?

It is true that stronger intellectual property rights (patents) came into biology with the rise of biotechnology.  However, the debate about the public policy about intellectual property rights in plants and animals, while a very important and highly-debated public policy, is separate and distinct from the regulatory system applied by governments to agricultural biotechnology.  The latter (regulatory policy) affects development and application more than the former (intellectual property rights).
The Golden rice story suggests otherwise, see: https://pubmed.ncbi.nlm.nih.gov/25437240/
This story indicates that the patent issues can be resolved fairly quickly and easily.  Hence, the hurdle for this humanitarian agricultural biotechnology has not been patents; the hurdle has been the excessive and disproportionate burdens from regulatory systems (both domestic laws/regulations and international conventions).
If the regulatory systems had been sensible, the participation of breeders in agricultural biotechnology with patents would have been basically equivalent to the participation of breeders in agricultural breeding with breeder’s rights.  The author describes agricultural breeding with breeder’s rights as “a very diverse plant breeding sector.”   I think the same description would apply to breeders in agricultural biotechnology with patents if the regulatory system had been sensible.  Thus, the authors are blaming “patents” for something that “patents” did not cause (at least not to any appreciable additional effect).  The cause of a non-diverse breeding sector in agricultural biotechnology is regulatory restraints, not intellectual property rights.

4th) there are relevant mistakes when reporting technical detals.
Two examples of this:
- at line 110-111: "Targeted mutagenesis: the targeted cutting and/or replacement of base pairs (TALEN, ODM etc. 2005) and techniques based on CRISPR-cas (2014)."
CRISPR-Cas is also a method essentially based on the precise cutting of the DNA, the main difference with TALEN and ZFN is that target recognition is mediated by an RNA molecule rather than a protein. On the other side ODM does not, to my knowledge, require double strand breaks.
Moreover targeting depents on the cutting. Outcome depends on the availability of a substrate for repair, but this issue is the same for all three techniques

- at line 197-198: "...targeted mutagenesis, unlike transgenesis, does not lead to plants that cannot develop spontaneously in nature".
There are plenty of examples of natural transgene among plant, microbial, insect and fungi. For instance whole bacterial genomes can be found integrated into insects or bacterial genes in plants or fungal genes in aphids (e.g.: https://science.sciencemag.org/content/328/5978/624 ). See also the sweet potato genome with bacterial genes in it.

- Line 153-155. "Transferring a gene from an unrelated species using SDN-3 produces plants that could not logically arise in nature."
Same point as the preceding one. This apparently true statement is not acceptable in reality: horizontal gene transfer is widespread in nature. Many examples at many levels in the biological world

5th) Some sentences are obscure/lame or seem to mix different levels (16-17)(124-126), (362-364)
For instance, (line 16-17): "This results in a generally positive approach with several arguments of rejection of gene editing"
Does that mean that the people rejecting gene editing have sound arguments? Make sentence clearer.

6th) The authors confuse things that must be treated separately: biodiversity within crop species (agrobiodiversity)  must not be confused with biodiversity at large (number of different species and their abundance in a certain environment) (see also below)

Other point of criticism (some times related to the issues raised above)

* (Line 160-162). Consequentialism:
...ethically correct action is characterized by a good result of the actions
The outcome depends on the choice of the focus. You decided to focus on "the (potential and probable) consequences with regard to sustainability, biodiversity, and potential negative effects derived from unintended side-effects of the technology."
If you consider however feeding humans with a healthy diet, then the outcome might be very different...

* Biodiversity (Line 212 -213)
"It mainly concerns three system levels: diversity in ecosystems (landscape), species diversity within ecosystems and genetic diversity within the species (within and between plant varieties)."
The ecosystems level has little to do with breeding.
Diversity within ecosystems: do they mean all species? Do they refer to cultivated fields? Do they refer to plant species only? Usually farmers want a single species in a field and namely the crop!
Genetic diversity within the species (within and between plant varieties).
Do they refer to the field level or as a whole (the sum of all varieties for that species)? Usually, agrobiodiversity in fields is minimal (varieties must be uniform to be identified and commercialized)

* Line 246: 4.3. Off-target effects
When crossing plants to add a positive trait to a variety, it often happens that in the progeny plants are found in which undesirable traits occur as well.
The authors fail to recognize that this is a major character that comes from domestication, see:
https://www.nature.com/articles/s41598-018-31041-0

* (line 222-223) "it is clear that the genetic diversity in Europe is increasing as a result of modern  breeding [16-18]. This is attributed to the introduction of breeder’s rights intensifying investment in breeding programmes in the 1970s, the use of marker-assisted selection".
If breeders right increase genetic diversity, the same can be argued fot transgenesis. There are now in cultivation in Spain and Portugal more than 100 Bt maize varieties, for instance.

* "...debates have taken place in society about the ethical acceptance of biotechnology in animal breeding for agriculture and horticulture."
Unclear

* "whether it would be unethical not (!) to use technical possibilities in breeding when they may tackle morally important issues such as hunger and climate change."
The exclamation mark suggests that you disagree. Please, make it clear and state your reasons for it.

* Line 105. "Transgenesis: the transfer of functional genes from one species to another (1970);"
The date is referring to the procedure in bacteria. If you refer to plants, 1982 is a better date.

* (Line 189-190) An example is herbicide tolerance, which allows for large-scale cultivation because less labour is required for weeding.

Line 195 GMO techniques
"GMO technique" is a nonsense expression, since it means Genetically Modified Organism techniques.
I understand many people, even professionals use it, but it is not acceptable, it has really no meaning same for "GMO cultivar" (Line 201), GMO regulation (Line 202)...

Line 195-196.  "It is therefore logical to realize the desire to reduce the use of herbicides through crop protection legislation, and not through biotechnology regulations."
Unclear

Minor points

Figure 1
I would rather call this a table...
Second point: the authors mention reference [1] as an "integrated social-ethical framework drawn up in 2002 for transgenesis" to describe figure 1. Point is that such reference is a Dutch text, therefore unavailable to the great majority of readers

Line 30: See also [2 [3]
line 166 It attempt (s missing)
Line 238-239 Gene banks, such as the Center for Genetic Resources in Wageningen. Lame sentence

References:
line 514-515. The link is broken and doe not work. Moreover the text is in Dutch, therefore I don't see it appropriate that the title in the reference is in English

Line 524-525. Link broken, it does not work

Line 587-588 Same as before

Figure 2
part of text is not visible, but is concealed in text boxes. For instance: "Which values
might be challenged" "Integral consider". One text is misspelled: integriteit of animals

Author Response

Thank you so much for your critical review. Your comments proved very valuable and we have accepted almost all. Some replies to your comments

  • You are right that we have not been overly clear about our choice of the ethical concepts. We have included a small paragraph on that and removed the figure 1 that we agree was not really necessary.
  • Yes we included a sentence clarifying the basic function of agriculture – to support human interests in food, feed, fibre etc.
  • The IP issue is definitely part of the social justice discussion just (like Farmers’ Rights and the current discussions on DSI in the frame of the CBD, which we added), even if they originate in different policy areas, and result in very different regulatory arrangements. They indeed do have a direct link with the emergence of plant biotechnologies (because of patents and DSI specifically focus on biotech). We agree that regulatory restraints are a major bottleneck, but the opposition to the trait-patents by almost all family owned companies (Netherlands – EU – global in ISF) as opposed to breeder’s rights shows that they see the former as a seriously restrictive.
  • Ref: ODM in relation to CRISPR and TALEN: sentence adapted
  • Ref transgenesis in nature: sentence added
  • Biodiversity; I don’t think we mix up the three main system levels (genetic, species, and landscape diversity, but your comment made us formulate this section much clearer with respect to the systems levels
  • Consequentialism: it is true that “good results of the actions” is necessarily value driven. We have added a section at the beginning introducing the basic roles of agriculture (to feed us etc.), and the importance of efficient production. We however keep our focus on the sustainability aspects – first, because policies currently stress that value, and secondly, because those are needed to feed us tomorrow as well . . . .
  • Thank you for the link to the article on cabbage domestication. Useful reference for some of my other work as well!
  • Breeder’s rights increasing diversity – this is claimed by the authors that we refer to. Their argument: breeder’s rights create more investments for breeding and thus more opportunities to use more distant germplasm which is costly compared with just crossing related varieties to breed a better one.
  • The next bullets and ‘minor points’ all accepted with Thanks!!!

So in summary - we have made some pretty substantial changes to the document.

Reviewer 2 Report

Comments

Obviously, after one rapid read, I cannot leave from the text. I have no matter to be identified in the review, I am, I hope, rational, not neutral : neither against biotechnologies (that I practiced since the 70’s) nor in favor of them. One aspect in breeding is lacking and it could be determinant to the ethical aspect.

1= Breeding is not made because of consumer requirements. Breeding ideas start after secrete meetings -in public and private - and only once the final variety is obtained the institute or the company should get authorization to release the variety as a commercial product; consumers haven’t the opportunity to pronounce on the methods used to get them before its commercial uses.

There are many filters to authorise the release: It was the job of specialists to control varieties.
2= Consumers are not involved at these levels.

3=Then the ‘biotech’ appeared enabling genome changes as described here. I accept the classification presented here.
4=Dealing with sustainability of the new traits the trouble is whether the new changes will prevent conservation of the collections and of each genoytpe. Every breeder knows that each trait to be conserved has to undergone a selection pressure. Consequently, when diseases become epidemically spread in the entire world losses in diversity inevitably occurred in collections – susceptible old varieties may disappear, every species are concerned, this is called erosion in genetic resources.

In my opinion when the new genotype is obtained – whatever the technologies – if it impairs genetic resources it should be prevented of diffusion, the trait could be invasive. This did not concern herbicide and abiotic resistant traits that are rapidly loss in population in the absence of treatments or selection pressure, but is very important to newly created disease resistance genes. When new races or new pathogen spread, it could be inevitable to loss ancient landscape cultivars, that reduces the future breeding potential.

I have made comments in the text to clear some sentences and to suggest some improvements.

5= as you noticed in the text L 49 after several years of work Breeders may more or less distort the history of their new varieties, and tow to prove they have did that?

Dating the events in biotechnology progresses is a good initiative, however i suggest to give concrete examples, I suggested some

I provide to you the odt file I have get from the pdf of your submission

Author Response

Dear Reviewer 1. Thank you so much for the detailed comments on our paper. They were very valuable and we have accepted almost all. Some comments on your general remarks:

1/2: The role of the consumer in plant breeding. It is our experience that breeders increasingly consult consumers. Obviously, this is most prominent in vegetables and flowers, where next to the interests of farmers and transporters also consumption qualities have created major breeding goals, such as the size of cauliflower (smaller families); taste (brix) of tomatoes, winterhardiness of gerbera making it a garden plant rather than just a cut flower, etc. One could also consider breeding for low-input organic agriculture a consequence of consumer demands. It is true that consumers are not involved in variety release, but they ‘vote’ through their purchasing power.

3: thank you; it is difficult to get the right literature references for all (especially the older) breeding methods; the references that you mentioned were most useful

4: you are right that this aspect (introgression of traits in natural populations in centres of origin/diversity) was not included. We did include it in the section on ‘biodiversity’.

One thing we didn’t adopt (but we did change the sentence to increase clarity) is our assessment that breeders cannot know the full history of their varieties when they use (under the breeding exemption) materials from other breeders, from other countries and possibly directly from collecting in nature or local markets. Breeders of some crops keep limited administration on pedigrees (eg flower breeding) and for others (both in private companies and institutes), breeding books may go back some decades, but when I was asking about the inclusion of the mutagenesis products on barley in the 1950s, there were no records going back that far. So even if breeders are serious about tracing back all materials in the pedigrees, they will have limitations.

But in conclusion:  your comments have improved the paper, so thank you very much !! We made some substial changes - including adding quite some references as you indicated.

Niels Louwaars & Henk Jochemsen

Round 2

Reviewer 1 Report

I acknowledge that the authors improved the text in a substantial manner. They also took care in integrating most of the critiques. Please find below additional comments to the authors. These comments mostly concern  sentences after revision.

I am still a bit troubled by some choices (for instance is landscape diversity really a sort of biodiversity? I would say it is not), but this sort of issues is more related to wording and philosophy and I would not ague further on those. However, on some points I believe the authors must get the record straight and avoid as much as possible wrong descriptions or definitions. A clear example of this is on horizontal gene transfer or on genome editing strategies (see below). Another example is the definition of transgenic. I am aware that these are technical matters that are difficult to be defined, especially if you are not in the business. The authors need not to explain everything. However, it is important to avoid mistakes or to mislead readers. Please find below a list of newly raised points on the revised manuscript. In general, they are on a minor level than those identified in the first round of revision.

- Introduction of diversity from other regions and crossing with the local genepool (from ca 1800 [11]);

Modern strawberries were created by Duchesne around 1760 in France by crossing a European species with a species from Chile: F. moschata and F. chiloensis

- Use of hybrid vigour by creating hybrids using controlled crossing of selected parent lines (1925 [12]) ;

I believe crossing of distant varieties and species is much older. For instance, triticale was first produced in 1873 by Scottish botanist A. S. Wilson.

- Transgenesis: the transfer of functional genes from one species to another (1980s [19]);

Not quite. A so-called cisgenic plant would still be considered transgenic according to regulatory authorities and to most scientists because the DNA was isolated and then introduced by transformation into the plant, regardless of the DNA origin. It is not the source of the genes, but rather the act of introducing the gene into the cell that is crucial. Present day legal framework focus much more on the method than on the source of the gene...

By the way, a transgenic (or cisgenic) plant may harbour a non-functional gene, but still be transgenic (or cisgenic).

- Targeted mutagenesis: the targeted cutting and/or replacement of base pairs

ODM would fit to a good degree with this definition (replacement of base pairs)

-Molecular Marker assisting selection

Molecular Marker assisted selection

-Gene editing (or genome editing) is the targeted modification of the DNA to change or disable the function of a gene

Then SDN-3 would not be included in such a definition, as it implies the addition of a new gene, not the change of a pre-existing one....

- SDN-1 or SDN-2 techniques, leading to small changes in the genome without crossing species boundaries, do not result in plants that could not also be obtained with traditional - widely accepted - breeding methods or that could not arise spontaneously in nature.

While I agree that in most cases SDN-2 is or shall be used to move genes across similar and compatible species, one could still use genes from distant species but which retain a significant sequence similarity or just use bordering sequence with sufficient sequence similarity. The key difference between SDN-2 and SDN-3, as far as I understand, is whether the donor DNA is already present as a similar sequence and a similar position in the genome or not.

- horizontal DNA transfer can occur from certain bacteria to their host plants.

HGT is a widespread phenomenon taking place among all life kingdoms and in both directions, not just among bacteria or from bacteria to eukaryotes, even though it is more frequent that the donor is a bacterium or a virus. See for instance:

https://www.mdpi.com/2223-7747/9/3/305/htm

https://www.nature.com/articles/ncomms2148

https://nph.onlinelibrary.wiley.com/doi/10.1111/nph.16022

https://www.cell.com/cell/fulltext/S0092-8674(21)00164-1

I am not claiming nor requesting that the authors should cite all of these papers or turn the manuscript into a review on horizontal gene transfer, not at all. I am just requesting that if you put a statement on HGT, then the statement must be correct and not just a sentence with some 'scientific' terms...

More generally, from the text it seems that SDN-1 or 2 are more acceptable than SDN-3. As a scientist, I strongly believe that the acceptability of a gene transfer must be based on the characteristics and the benefits/risks of the product obtained. One could transfer an almond gene back into almond (via breeding or transgenesis or genome editing) and yet create a toxic plant capable of killing people. On the other side transferring a bacterial gene (coding for instance for a Bt toxin or a carotenoid synthesizing enzyme) might imply minute risks and large benefits (lower mycotoxins or better nutrition). The first transfer would be unethical in my worldview, but not the second.

-shelf life of flowers, vegetables and fruits (reduction of food losses)

If you refer also to flowers, then it is not just food

-The actual contributions of breeding to sustainability is under discussion. Breeding can adapt crops to certain cultivation techniques, whose contribution to sustainability is debated. An example is herbicide tolerance, which allows for large-scale cultivation because less labour is required for weeding, but which increases the use of chemicals with an impact on the environment.

This is a good example of bias, IMHO, that needs to be corrected. Even hand weeding has got a strong environmental impact, as well as a strong impact on the health of workers. Should you try to spend one day long to perform hand weeding over a hectare of land, you would perceive the importance of mechanized or chemical weeding. If you want to judge the goodness of a crop or of a technology on the consequences, then you should be considering all the consequences, not just a purported and undefined ill effect on the environment, but also those on the life and the health of people using or not using that technology.

-Furthermore, although the most widely debated form ("Roundup Ready") of herbicide tolerance is a result of biotechnology, it can be pointed out that also non-GM plants exist that are tolerant to certain weed killers.

I acknowledge that the authors improved this part in a substantial manner. I wonder why they are not questioning that the opposition of the environmental movements is almost only concentrated on the biotech versions, not the other ones, that are widespread in Europe (e.g. rice and sunflower). IMHO, this betrays ignorance and cherry picking (of the environmental movement). I also wonder why the authors fail to address the responsibility of the environmental movement in raising the regulatory requests on the technology and therefore of its costs, thereby collaborating with the seed/chemical multinationals in reducing accessibility of this technology to small enterprises (this is just a comment, not a request). It would be perfect for the section ‘Social justice in hindsight‘…

-GMO cultivar.

Not a sensible expression, as already pointed out before. A cultivar already refers to an organism. Saying GMO cultivar equals to saying Genetically modified organism cultivar. On the GMO concept, see: https://pubmed.ncbi.nlm.nih.gov/26348954/  https://pubmed.ncbi.nlm.nih.gov/26559524/ and related papers from the same author. ‘GMO regulation’ (or ‘GMO trait’) may be more acceptable because could be interpreted as the regulation of GMOs.

-This Centre for Genetic Resources in Wageningen thus saves a lot of labour and greenhouse space.

The Center was not mentioned before, so "this" seems inappropriate to me.

(wheat multi-line TUMULT [41]

missing closing bracket

-random mutagenesis, lead to many more side effects or lucky positives, than targeted mutagenesis via CRISPR Cas.

I believe the authors probably meant: …random mutagenesis, lead to many more side effects or to fewer lucky positives, than targeted mutagenesis via CRISPR/Cas.

-in relation to novel breeding methods are indeed very significant when applied to humans or animals

I would not speak of breeding methods referring to human beings...

-Next to these legal changes. FRAND

The sentence presumably contains a mistake and should read: Next to these legal changes, FRAND

-Intellectual property Rights

Intellectual Property Rights

-Gene editing is technically simple and cheap to apply compared to transgenesis

Transgenesis is also quite technically simple and cheap to apply. Indeed, most genome editing requires, as an obligatory intermediate step, a transgenic plant. Therefore, the statement sounds partially contradictory.

-Here we mainly focus on rights (as legitimate moral claims and intrinsic value.

missing closing bracket

-The right of the plant to fulfil its natural purpose and to be treated as an autonomous-self-regulating being (integrity of life);

This statement (from Lammerts van Bueren and Struik) is contradictory when applied to domesticated plants and animals. Most of them are unable to survive in the absence of human care. The manuscript correctly discusses this in reference to b), but fails to recognize it in respect to a). As far as d) is concerned, it would be necessary to define "the nature of the plant", but this is something beyond the scope of this manuscript.

-Utz for coffee

unclear reference (to me)

-However, presenting on every product all breeding methods that have been used, puts a significant strain on processors (...) and the consumer

I'd say on the whole chain starting from the breeder and seed companies, growers... Who is going to pay for the generation and the maintenance of such information, which is completely useless, annoying or irrelevant to most consumers? That is also a relevant ethical issue!

Author Response

Dear Reviewer,

Thank you very much for going through our paper in such detail. We have adapted the paper on most issues - below we explain our actios based on all your comments. Once more, we appreciate your contribution to improving the paper.

kind regards Niels Louwaars & Henk Jochemsen

- Introduction of diversity from other regions and crossing with the local genepool (from ca 1800 [11]);

Modern strawberries were created by Duchesne around 1760 in France by crossing a European species with a species from Chile: F. moschata and F. chiloensis

We fully agree that explicit crossing (with plants from other regions) has been quite common before 1800 in horticulture (including fruit trees and flowers), but to our knowledge – what we found in literature - not in the major field crops. Crossing French with Ukrainian wheats created a significant improvement in France early 19th century.

- Use of hybrid vigour by creating hybrids using controlled crossing of selected parent lines (1925 [12]) ;

I believe crossing of distant varieties and species is much older. For instance, triticale was first produced in 1873 by Scottish botanist A. S. Wilson.

 Here we refer to hybrid varieties (of maize in the USA) – not interspecific crosses

- Transgenesis: the transfer of functional genes from one species to another (1980s [19]);

Not quite. A so-called cisgenic plant would still be considered transgenic according to regulatory authorities and to most scientists because the DNA was isolated and then introduced by transformation into the plant, regardless of the DNA origin. It is not the source of the genes, but rather the act of introducing the gene into the cell that is crucial. Present day legal framework focus much more on the method than on the source of the gene...

Both in the USA (and in the recent EU report and earlier EFSA assessments, cis-genesis is treated separately from transgenesis, even though they use exactly the same techniqus. In the Netherlands, Parliament made that explicit distinction back in 2014, when on grounds of ethics (not crossing species barriers) legislators were called to ‘free’ cis-genesis from the GMO-laws. 

By the way, a transgenic (or cisgenic) plant may harbour a non-functional gene, but still be transgenic (or cisgenic).

AGREE, but practically, the technology targets functional genes.

- Targeted mutagenesis: the targeted cutting and/or replacement of base pairs

ODM would fit to a good degree with this definition (replacement of base pairs)

 The other reviewer suggested to treat ODM slightly differently because it does not involve double stranded cuts

-Molecular Marker assisting selection

Molecular Marker assisted selection

Thank you – indeed a typo

-Gene editing (or genome editing) is the targeted modification of the DNA to change or disable the function of a gene

Then SDN-3 would not be included in such a definition, as it implies the addition of a new gene, not the change of a pre-existing one....

Agree – we have reformulated this 

- SDN-1 or SDN-2 techniques, leading to small changes in the genome without crossing species boundaries, do not result in plants that could not also be obtained with traditional - widely accepted - breeding methods or that could not arise spontaneously in nature.

While I agree that in most cases SDN-2 is or shall be used to move genes across similar and compatible species, one could still use genes from distant species but which retain a significant sequence similarity or just use bordering sequence with sufficient sequence similarity. The key difference between SDN-2 and SDN-3, as far as I understand, is whether the donor DNA is already present as a similar sequence and a similar position in the genome or not.

Indeed in SDN-2 a template can be introduced that is based on allelic information from (homologuous) gene from a more or less related species. However, in SDN-2 you normally do not use the foreign gene itself, but just the sequence information.

- horizontal DNA transfer can occur from certain bacteria to their host plants.

HGT is a widespread phenomenon taking place among all life kingdoms and in both directions, not just among bacteria or from bacteria to eukaryotes, even though it is more frequent that the donor is a bacterium or a virus. See for instance:

https://www.mdpi.com/2223-7747/9/3/305/htm

https://www.nature.com/articles/ncomms2148

https://nph.onlinelibrary.wiley.com/doi/10.1111/nph.16022

https://www.cell.com/cell/fulltext/S0092-8674(21)00164-1  : Here, we show that, through an exceptional horizontal gene transfer event, the whitefly has acquired

I am not claiming nor requesting that the authors should cite all of these papers or turn the manuscript into a review on horizontal gene transfer, not at all. I am just requesting that if you put a statement on HGT, then the statement must be correct and not just a sentence with some 'scientific' terms...

Thank you – we have re-formulated the sentence and added one of your references

More generally, from the text it seems that SDN-1 or 2 are more acceptable than SDN-3. As a scientist, I strongly believe that the acceptability of a gene transfer must be based on the characteristics and the benefits/risks of the product obtained. One could transfer an almond gene back into almond (via breeding or transgenesis or genome editing) and yet create a toxic plant capable of killing people. On the other side transferring a bacterial gene (coding for instance for a Bt toxin or a carotenoid synthesizing enzyme) might imply minute risks and large benefits (lower mycotoxins or better nutrition). The first transfer would be unethical in my worldview, but not the second.

From a plant-science point of view we can certainly agree. However, almost all regulatory systems that we know (with the exception of Canada) are initially process based. Several countries, but most notably Japan may not regulate plants derived from SDN-1; have very minor (information-) obligations on SDN-2 and consider SDN-3 just like the older – commonly not location specific transgenes. Similarly, from various ethical angles, SDN-1 (and simple SDN-s) can be considered to occur in accepted plant breeding (random mutagenesis), whereas SDN-3 – even if it might occur in nature as you rightfully say (HGT) - is not used in conventional breeding.

-shelf life of flowers, vegetables and fruits (reduction of food losses)

If you refer also to flowers, then it is not just food

You are indeed right – adjusted the sentence (even though flowers are increasingly used by chefs J )

-The actual contributions of breeding to sustainability is under discussion. Breeding can adapt crops to certain cultivation techniques, whose contribution to sustainability is debated. An example is herbicide tolerance, which allows for large-scale cultivation because less labour is required for weeding, but which increases the use of chemicals with an impact on the environment.

This is a good example of bias, IMHO, that needs to be corrected. Even hand weeding has got a strong environmental impact, as well as a strong impact on the health of workers. Should you try to spend one day long to perform hand weeding over a hectare of land, you would perceive the importance of mechanized or chemical weeding. If you want to judge the goodness of a crop or of a technology on the consequences, then you should be considering all the consequences, not just a purported and undefined ill effect on the environment, but also those on the life and the health of people using or not using that technology.

Even though we agree with you, the issue of large scale herbicide use is (whether we like it or not) a strongly debated issue in society – as can be read from court cases in Franec that even went up to the European Court of Justice. We don’t dig into this discussion any deeper, because that would take this paper too far. We have adjusted the sentence though.

-Furthermore, although the most widely debated form ("Roundup Ready") of herbicide tolerance is a result of biotechnology, it can be pointed out that also non-GM plants exist that are tolerant to certain weed killers.

I acknowledge that the authors improved this part in a substantial manner. I wonder why they are not questioning that the opposition of the environmental movements is almost only concentrated on the biotech versions, not the other ones, that are widespread in Europe (e.g. rice and sunflower). IMHO, this betrays ignorance and cherry picking (of the environmental movement). I also wonder why the authors fail to address the responsibility of the environmental movement in raising the regulatory requests on the technology and therefore of its costs, thereby collaborating with the seed/chemical multinationals in reducing accessibility of this technology to small enterprises (this is just a comment, not a request). It would be perfect for the section ‘Social justice in hindsight‘…

See our remark on you rprevious remark. Maybe a topic for a next paper?? Actually, I wrote a brief note in our university paper in the late 1990s under the heading “Greenpeace supports Monsanto” with exactly your line of thinking. The environmental opposition to GMOs have removed all competitors (public and private) from the market that the multinational  companies dominate!!!

-GMO cultivar.

Not a sensible expression, as already pointed out before. A cultivar already refers to an organism. Saying GMO cultivar equals to saying Genetically modified organism cultivar. On the GMO concept, see: https://pubmed.ncbi.nlm.nih.gov/26348954/  https://pubmed.ncbi.nlm.nih.gov/26559524/ and related papers from the same author. ‘GMO regulation’ (or ‘GMO trait’) may be more acceptable because could be interpreted as the regulation of GMOs.

Agree – I though I had corrected that - sorry

-This Centre for Genetic Resources in Wageningen thus saves a lot of labour and greenhouse space.

The Center was not mentioned before, so "this" seems inappropriate to me.

 Sentnec adjusted

(wheat multi-line TUMULT [41]

missing closing bracket

 thank you

-random mutagenesis, lead to many more side effects or lucky positives, than targeted mutagenesis via CRISPR Cas.

I believe the authors probably meant: …random mutagenesis, lead to many more side effects or to fewer lucky positives, than targeted mutagenesis via CRISPR/Cas.

 An unintended side effect can be negative but also be a ‘lucky positive’. We have clarified the sentence  

-in relation to novel breeding methods are indeed very significant when applied to humans or animals

I would not speak of breeding methods referring to human beings...

 Agree – changednovel breeding methods into ‘gene editing’

-Next to these legal changes. FRAND

The sentence presumably contains a mistake and should read: Next to these legal changes, FRAND

 Indeed, thanks

-Intellectual property Rights

Intellectual Property Rights

 Thank you

-Gene editing is technically simple and cheap to apply compared to transgenesis

Transgenesis is also quite technically simple and cheap to apply. Indeed, most genome editing requires, as an obligatory intermediate step, a transgenic plant. Therefore, the statement sounds partially contradictory.

This is indeed a strongly debated issue, relating to the legal definition of ‘transgenic’. I hope it is clear that in this context ‘transgenic’ is used in the meaning that a functional gene (from a different species) has been introduced into the cell, and has shown to perform its function.

We understand that technically (and when sufficient genomic information is available); gene editing is quite simple, and especially the selection processes after regeneration are much simpler because of the targeted approach, where in conventional transgenesis the many different products (where and in how many copies has the transgene been introduced, requires much more work. Of course the costs of the two techniques also relate to licensing costs etc, but we refer here to the technical costs only.

-Here we mainly focus on rights (as legitimate moral claims and intrinsic value.

missing closing bracket

Thank you

-The right of the plant to fulfil its natural purpose and to be treated as an autonomous-self-regulating being (integrity of life);

This statement (from Lammerts van Bueren and Struik) is contradictory when applied to domesticated plants and animals. Most of them are unable to survive in the absence of human care. The manuscript correctly discusses this in reference to b), but fails to recognize it in respect to a). As far as d) is concerned, it would be necessary to define "the nature of the plant", but this is something beyond the scope of this manuscript.

Here again, we may agree with you, but this is the clearest reference to the ethical debate on these issues, and their article has been very influential in the creation of a negative position of IFOAM, the international organic agriculture organisation. Irrespective of whether we agree or not – we consider it essential to refer to this paper. (and I think we do discuss it quite critically)

-Utz for coffee

unclear reference (to me)

Utz is the widely used certification mark for ‘sustainably produced’ coffee. Utz is now part of Rainforest Alliance, which has a broader scope, is thus more widely known and easily found with google when readers are interested.

-However, presenting on every product all breeding methods that have been used, puts a significant strain on processors (...) and the consumer

I'd say on the whole chain starting from the breeder and seed companies, growers... Who is going to pay for the generation and the maintenance of such information, which is completely useless, annoying or irrelevant to most consumers? That is also a relevant ethical issue!

We adapted the sentence to include your valid and question that is indeed relevant for this paper!
